

# Convex-hull mass estimates of the dodo (*Raphus cucullatus*): application of a CT-based mass estimation technique

Charlotte A. Brassey[1], Thomas G. O'Mahoney[1], Andrew C. Kitchener[2,3], Phillip L. Manning[4,5] and William I. Sellers[1]

[1] Faculty of Life Sciences, University of Manchester, Manchester, United Kingdom
[2] Department of Natural Sciences, National Museum of Scotland, Edinburgh, United Kingdom
[3] School of Geosciences, University of Edinburgh, Edinburgh, United Kingdom
[4] Interdisciplinary Centre for Ancient Life, School of Earth, Atmospheric and Environmental Sciences, University of Manchester, Manchester, United Kingdom
[5] Department of Geology and Environmental Geosciences, College of Charleston, Charleston, SC, United States of America

Corresponding author
Charlotte A. Brassey,
charlotte.brassey@manchester.ac.uk

## ABSTRACT

The external appearance of the dodo (*Raphus cucullatus,* Linnaeus, 1758) has been a source of considerable intrigue, as contemporaneous accounts or depictions are rare. The body mass of the dodo has been particularly contentious, with the flightless pigeon alternatively reconstructed as slim or fat depending upon the skeletal metric used as the basis for mass prediction. Resolving this dichotomy and obtaining a reliable estimate for mass is essential before future analyses regarding dodo life history, physiology or biomechanics can be conducted. Previous mass estimates of the dodo have relied upon predictive equations based upon hind limb dimensions of extant pigeons. Yet the hind limb proportions of dodo have been found to differ considerably from those of their modern relatives, particularly with regards to midshaft diameter. Therefore, application of predictive equations to unusually robust fossil skeletal elements may bias mass estimates. We present a whole-body computed tomography (CT) -based mass estimation technique for application to the dodo. We generate 3D volumetric renders of the articulated skeletons of 20 species of extant pigeons, and wrap minimum-fit 'convex hulls' around their bony extremities. Convex hull volume is subsequently regressed against mass to generate predictive models based upon whole skeletons. Our best-performing predictive model is characterized by high correlation coefficients and low mean squared error ($a = -2.31$, $b = 0.90$, $r^2 = 0.97$, MSE = 0.0046). When applied to articulated composite skeletons of the dodo (National Museums Scotland, NMS.Z.1993.13; Natural History Museum, NHMUK A.9040 and S/1988.50.1), we estimate eviscerated body masses of 8–10.8 kg. When accounting for missing soft tissues, this may equate to live masses of 10.6–14.3 kg. Mass predictions presented here overlap at the lower end of those previously published, and support recent suggestions of a relatively slim dodo. CT-based reconstructions provide a means of objectively estimating mass and body segment properties of extinct species using whole articulated skeletons.

## INTRODUCTION

Body mass ($M_b$) is a fundamental descriptor of an organism and co-varies with important ecological and physiological traits, such as population density, metabolism and cost-of-transport (*Schmidt-Nielsen, 1984*). Key evolutionary scenarios, such as the origin of avian flight (*Turner et al., 2007*) and the extinction of island flightless avian species (*Boyer, 2008*), have been diagnosed on the basis of estimated $M_b$. Therefore, the reconstruction of body mass in extinct bird species is a subject of considerable interest within the palaeontological and evolutionary biology literature (*Turner et al., 2007*; *Boyer, 2008*; *Hone et al., 2008*; *Butler & Goswami, 2008*; *Brassey et al., 2013*).

An often-applied technique for estimating the body mass of an extinct vertebrate has been to measure a skeletal dimension from modern species, such as femur circumference (*Campione & Evans, 2012*) or glenoid diameter (*Field et al., 2013*), and apply this as the independent variable in a regression against body mass. However, 'overdevelopment' of the pelvic apparatus has been found to be significantly correlated with the flightless condition in extant birds (*Cubo & Arthur, 2000*). Therefore, the application of mass prediction equations, based solely on hind limb material of flightless avian taxa, has been questioned in extinct species such as the moa (*Brassey et al., 2013*).

The dodo (*Raphus cucullatus*, *Linnaeus, 1758*) is an iconic representative of island flightlessness and human-induced extinction, and its external appearance has been a source of considerable intrigue due to the scarcity of trustworthy contemporaneous accounts or depictions (*Hume, 2006*). This extinct flightless columbiform was endemic to the island of Mauritius. However, the skeletal anatomy of the dodo is comparatively well known, and its pelvic morphology has been thoroughly investigated. Hind limb bones of *R. cucullatus* have been found to differ considerably in both their length and width relative to their volant relatives (*Hume, 2006*, although see *Livezey, 1993*). Yet previous attempts to estimate the body mass of the dodo have predominantly relied upon predictive equations derived solely from the hind limb metrics of extant species (*Angst, Buffetaut & Abourachid, 2011a*; *Louchart & Mourer-Chauviré, 2011*; *Angst, Buffetaut & Abourachid, 2011b*).

An alternative approach to mass estimation involves the reconstruction of 3D volumetric models. An early volumetric reconstruction of the dodo was conducted by physically sculpting a scale model of an individual and estimating volume via fluid displacement (*Kitchener, 1993*). Whilst such volumetric techniques are less liable to bias by individual robust/gracile postcranial elements than traditional linear bivariate equations, they do inevitably involve some degree of artistic licence in the sculpting of soft tissue contours and require an estimate for fossil body density to be assigned.

Following advances in 3D imaging technology, the use of digital skeletal models in mass estimation of fossil skeletons has become increasingly popular (*Seebacher, 2001*; *Hutchinson, Ng-Thow-Hing & Anderson, 2007*; *Gunga et al., 2008*; *Bates et al., 2009*). These studies typically involve the 'wrapping' of geometric shapes or lofted smooth surfaces around the skeleton in order to replicate the original soft tissue contour of the animal. Zero-density cavities such as lung and tracheal space may also be modeled (*Allen, Paxton & Hutchinson, 2009*). However, similar to physical sculpting with clay, assumptions

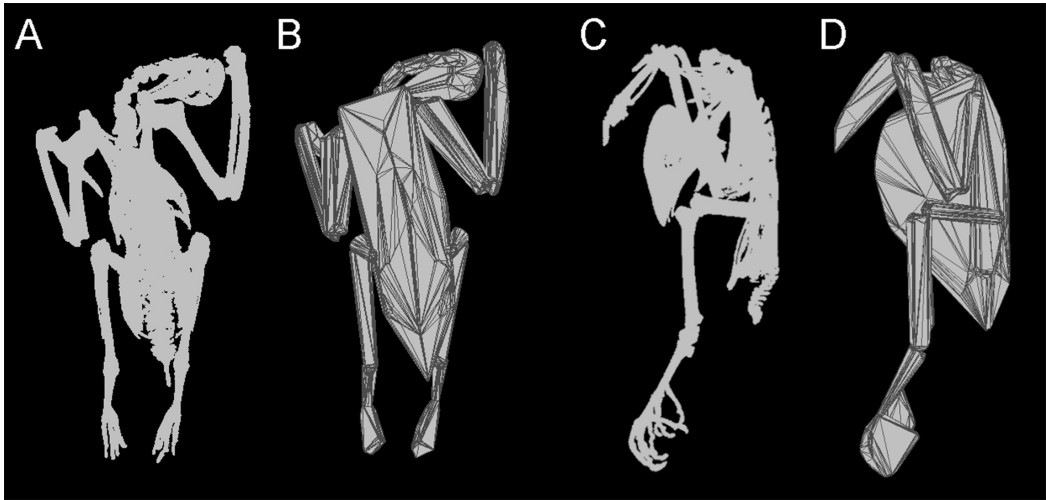

**Figure 1 Convex hulling process.** Example of the convex hulling process applied to the CT scanned carcass of a Victoria crowned pigeon (*Goura victoria*) from which the skeleton has been segmented. (A) and (C), skeleton in dorsal and lateral view respectively and; (B) and (D), corresponding convex hulls fitted to the functional units of the skeleton. Note convex hulls fitted to the feet in 1D are strongly influenced by the positioning of the toes (see in text for 'Discussion').

must still be made regarding body density and the extent of soft tissues beyond the skeleton. Therefore, it is essential that reconstructions are grounded within a quantitative understanding of extant species in order to avoid subjective modeling of soft tissues (both body and plumage).

Here we present new mass estimates for the dodo based on an alternative 'convex hull' volumetric reconstruction approach (*Sellers et al., 2012*; *Brassey & Sellers, 2014*). The convex hull (*CH*) of a set of points is defined as the smallest convex polytope that contains all said points, and intuitively can be thought of as a shrink-wrap fit around an object (see Fig. 1). Application of the convex hulling technique to mass estimation does not involve any subjective reconstruction of soft tissue anatomy and solely relies upon the underlying skeleton. We calculate minimum convex hull volumes for a sample of composite articulated dodo skeletons, and convert these to body mass estimates using a computed tomography (CT) calibration dataset of 20 species of extant pigeon. To our knowledge, this is the first time such an extensive CT dataset of extant animals has been used to reconstruct the body mass of a fossil of an extinct species.

## METHODS AND MATERIALS

The modern dataset consists of 20 columbiform individuals, spanning a wide variety of body sizes from a 70 g fruit dove (*Ptilinopus*; *Swainson, 1825*) up to the largest extant pigeon, the 2 kg Victoria crowned pigeon (*Goura victoria*; *Fraser, 1844*). We also cover a broad taxonomic range (including the closest extant relative of the dodo (*Shapiro et al., 2002*), the Nicobar pigeon (*Caloenas nicobarica*; *Linnaeus, 1758*)). Frozen carcasses were sourced from National Museums Scotland, Edinburgh, and the University of Manchester (see Table 1). Carcasses were CT scanned at Leahurst Veterinary School, University of

**Table 1 Specimen list of modern pigeons.** Specimen list of modern pigeons sourced from National Museums Scotland, Edinburgh. All other specimens were loaned from National Museums Scotland, Edinburgh.

| Species name | Common name | Body mass (g) | Preparation | $CH_{vol}$ (mm$^3$) | $CH_{vol}$—feet (mm$^3$) |
|---|---|---|---|---|---|
| *Goura victoria* | Victoria crowned pigeon | 1,951 | Eviscerated | 1,487,180 | 1,436,777 |
| *Streptopelia decaocto*[a] | Collared dove | 201 | Intact | 203,875 | 196,791 |
| *Columba livia* | Rock dove | 290 | Intact | 115,945 | 113,074 |
| *Columba palumbus*[a] | Wood pigeon | 305 | Intact | 337,993 | 328,279 |
| *Ducula bicolor* | Pied imperial pigeon | 450 | Eviscerated | 337,377 | 329,220 |
| *Petrophassa rufipennis* | Chestnut-quilled rock pigeon | 314 | Eviscerated | 303,511 | 286,104 |
| *Otidiphaps nobilis* | Pheasant pigeon | 401 | Eviscerated | 344,368 | 329,238 |
| *Nesoenas mayeri* | Pink pigeon | 200 | Eviscerated | 197,171 | 185,981 |
| *Ducula* sp. | Imperial pigeon | 336 | Eviscerated | 314,985 | 305,344 |
| *Caloenas nicobarica* | Nicobar pigeon | 539 | Eviscerated | 383,736 | 367,753 |
| *Phaps chalcoptera* | Common bronze-winged pigeon | 249 | Intact | 213,953 | 208,942 |
| *Ducula aenea* | Green imperial pigeon | 483 | Intact | 348,268 | 336,968 |
| *Columba guinea* | Speckled pigeon | 158 | Intact | 105,156 | 102,041 |
| *Zenaida graysoni* | Socorro dove | 176 | Eviscerated | 105,776 | 102,441 |
| *Gallicolumba* sp. | Bleeding heart dove | 215 | Eviscerated | 163,764 | 152,136 |
| *Phapitreron leucotis* | White-eared brown dove | 107 | Eviscerated | 69,424 | 67,088 |
| *Ptilinopus* sp. | Fruit dove | 71 | Eviscerated | 47,816 | 46,635 |
| *Ptilinopus superbus* | Superb fruit dove | 137 | Eviscerated | 77,882 | 7,4691 |
| *Treron vernans* | Pink-necked green pigeon | 167 | Eviscerated | 104,991 | 101,984 |
| *Ocyphaps lophotes* | Crested pigeon | 107 | Intact | 67,451 | 64,011 |

**Notes.**

$CH_{vol}$, minimum convex hull volume of the skeleton; $CH_{vol}$, feet, minimum convex hull volume minus the volume of the feet.

[a] Indicates specimens were sourced from the University of Manchester

Liverpool, in a Toshiba Aquilion PRIME helical scanner at a slice thickness of 0.5 mm and a pixel spacing of between 0.24–0.51 mm, depending on the maximum size of the specimen. 3D models of the skeletons were generated in Seg3D (*CIBC, 2014*), using an automatic threshold with subsequent manual masking to remove the dense rachises attached to the forelimb.

Models were exported into Geomagic Studio (www.geomagic.com), where each skeleton was divided into functional units (skull, neck, trunk, humerus, radius + ulna, carpometacarpals, femur, tibiotarsus + fibula, tarsometatarsus, feet). The cervical series was further subdivided in order to achieve a tight-fitting hull around the curving neck. Minimum convex hulls were calculated in MATLAB (www.mathworks.com), using the 'convhull' function implementing the Quickhull (qhull) algorithm (*Barber, Dobkin & Huhdanpaa, 1996*), and total convex hull volume was calculated as the sum of individual segment volumes (see Fig. 1). Body mass was measured for each carcass, and the relationship between $M_b$ and convex hull volume ($CH_{vol}$) was estimated using ordinary least squares (OLS) regression on $\log_{10}$ transformed data. As the purpose of the regression was to derive a predictive equation, a type-I regression, such as OLS, was deemed most appropriate (*Smith, 2009*). Additionally we accounted for the statistical non-independence of phylogenetically-related data points by carrying out phylogenetic generalized least

squares (PGLS) regressions, implemented in MATLAB using 'Regression2' software (*Lavin et al., 2008*). A majority-rule consensus tree was calculated using the R package 'ape' (*Paradis, Claude & Strimmer, 2004*) based upon a sample of 10,000 trees sourced from the birdtree.org website (*Jetz et al., 2012*) using the (*Hackett et al., 2008*) phylogeny as a backbone. All branch lengths were set to 1.

To reconstruct the body masses of articulated dodo skeletons, we generated 3D digital models of these specimens. It must be highlighted here that all dodo skeletons included in the present study are composites, likely comprising different individuals of varying age and/or sex (see below for further 'Discussion.' The Edinburgh dodo (National Museums Scotland, NMS.Z.1993.13) was scanned using a $Z + F$ Imager 5010 LiDAR (Light Detection And Range) scanner and reconstructed in the $Z + F$ LaserControl software. The Natural History Museum (NHMUK), London specimens (Tring skeleton, S/1988.50.1; South Kensington specimen NHM A9040) were digitized using the photogrammetric technique detailed elsewhere (*Falkingham, 2012*; *Mallison & Wings, 2014*) and reconstructed in VisualSFM (*Wu, 2011*). Despite application of two alternative imaging techniques, previous studies have found the results obtained via photogrammetry and laser scanning to be comparable (*Falkingham, 2012*), and convex hull results to be insensitive to point cloud density (*Brassey & Sellers, 2014*; *Brassey & Gardiner, 0000*). 3D models of the dodo skeletons were cleaned up in Geomagic and subdivided into functional units. Our only intervention with the dodo models was to mirror the right hand side of the Edinburgh ribcage to account for missing ribs on its left side. Convex hulls were fitted according to the methodology applied to modern pigeons.

The largest extant pigeon (*G. victoria*) weighs on average 2.3 kg (*Dunning, 1992*), a value far below all previous estimates of dodo mass. When applying a pigeon-based equation to predict dodo body mass, it is therefore necessary to extrapolate beyond the body size range upon which the predictive model is based. By restricting ourselves to phylogenetically closely related species, the fossil species of interest may therefore be up to an order of magnitude greater in size than any extant relative. Furthermore, the majority of modern pigeons included in this dataset are proficient fliers and have likely been subject to very different evolutionary pressures than the flightless dodo.

For this reason, we also applied a previously published convex hull equation derived from extant ratites and galloanserae birds, extending the range of body masses beyond 60 kg and incorporating ground-dwelling species. Raw data are taken from *Brassey & Sellers (2014)*, whilst the axes have been inverted ($\log_{10}$ volume as the independent variable vs. $\log_{10}$ mass as the dependent variable) to create a predictive model. Standard OLS regression was preferred as previous analyses found uncorrected type-I models to fit the data better than phylogenetically corrected regressions (*Brassey & Sellers, 2014*). It must be emphasized that the non-pigeon data are derived from an earlier study applying a different imaging technique (light detection and range, LiDAR, on museum mounted skeletons) and uses literature-assigned values for mass due to lack of associated body masses. Whilst the previous study found no significant impact on calculated $CH_{\mathrm{vol}}$ due

**Table 2 Details of mass prediction equations.** Ordinary least squares regressions of $\log_{10}$ body mass (g) against $\log_{10}$ convex hull volume ($CH_{vol}$, mm$^3$). Ground-dwelling refers to the predictive equation based upon ratites and fowl derived from *Brassey & Sellers (2014)*.

| Model | a | a (±95%) | b | b (±95%) | $r^2$ | p | AIC$_{OLS}$ | AIC$_{PGLS}$ |
|---|---|---|---|---|---|---|---|---|
| Eviscerated | −2.31 | −2.90−−1.72 | 0.89 | 0.78−1.00 | 0.97 | <0.001 | −28.42 | −20.97 |
| — minus feet | −2.31 | −2.87−−1.74 | 0.90 | 0.79−1.00 | 0.97 | <0.001 | −29.38 | −22.22 |
| Intact | −1.08 | −3.69−−1.53 | 0.66 | 0.16−1.16 | 0.70 | 0.019 | −5.29 | −10.15 |
| — minus feet | −1.06 | −3.62−−1.50 | 0.66 | 0.17−1.15 | 0.70 | 0.018 | −5.41 | −10.41 |
| Combined | −2.08 | −2.75−−1.42 | 0.85 | 0.72−0.98 | 0.92 | <0.001 | −34.42 | −26.64 |
| — minus feet | −2.08 | −2.73−−1.42 | 0.85 | 0.73−0.98 | 0.92 | <0.001 | −34.94 | −27.39 |
| Ground-dwelling | −1.65 | −2.52−−0.77 | 0.82 | 0.69−0.95 | 0.97 | <0.001 | | |

**Notes.**
±95%, 95% confidence intervals of the intercept and slope; MSE, mean square error of the regression; AIC, Akaike Information Criterion calculated for Ordinary Least Squares (OLS) and Phylogenetically Generalised Least Squares (PGLS).

to variation in point cloud density associated with different imaging techniques, caution should be exercised when comparing the regression models.

# RESULTS

Total convex hull volumes for the modern pigeons are reported in Table 1, and segment-specific $CH_{vol}$ values can be found in Supplemental Information 1. Convex hull models are available for download from http://www.animalsimulation.org. We found considerable variation between frozen pigeon specimens in the posture of the digits forming the foot i.e., adduction vs. abduction of the digits. This influenced the overall shape, and hence calculated $CH_{vol}$, of the foot functional units (see Fig. 1D). Given repositioning of the skeleton was not possible due to the frozen nature of the carcasses, here we report total $CH_{vol}$ values with and without feet included. External inspection of the carcasses suggested evisceration had been carried out on some specimens. Using CT scans the occurrence of evisceration was confirmed across our modern dataset (see Table 1). Therefore, we report separate predictive models derived from 'eviscerated' carcasses ($n = 13$), 'intact' carcasses ($n = 7$), and a third 'combined' model comprising both eviscerated and intact specimens ($n = 20$).

The results of the OLS regression analyses are presented in Table 2, and phylogenetically corrected (PGLS) regressions are given in Supplemental Information 1 alongside the composite phylogeny used in this analysis. PGLS regressions did not provide a better fit to the data than uncorrected OLS regressions (as determined by Akaike Information Criterion values, AIC) for the 'eviscerated' and 'combined' models (Table 2). However, a PGLS model was found to fit the 'intact' extant pigeon data better than an uncorrected OLS model (Table 2).

Removing $CH_{vol}$ of the feet from the analyses had very little effect on the results of the regression, although mean squared error (MSE) decreased slightly in all models and therefore only regression models minus feet are discussed any further in the text. Figure 2 illustrates a strong positive correlation between $M_b$ and $CH_{vol}$ for the eviscerated

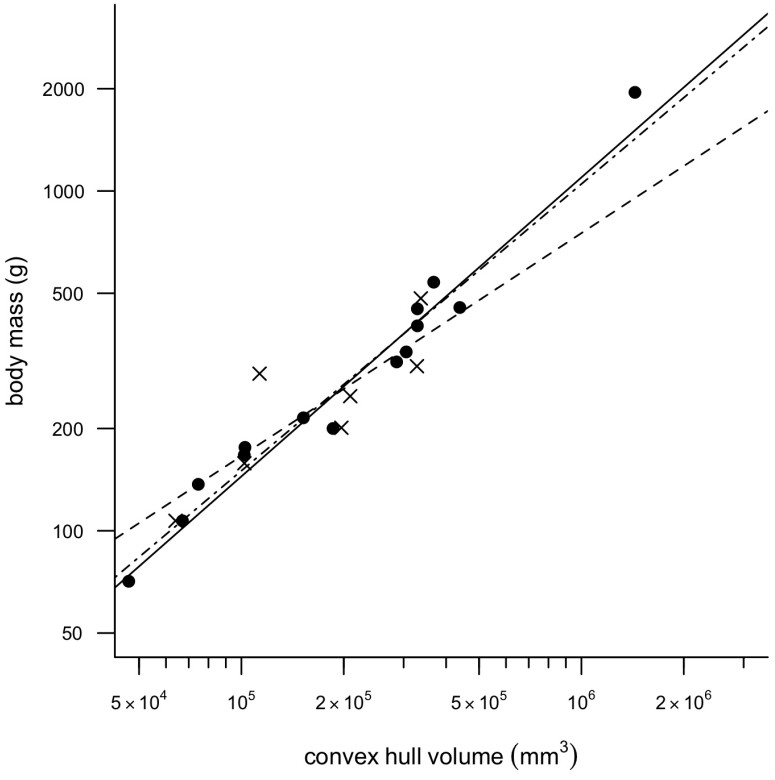

**Figure 2 Convex hull predictive model.** OLS regression results. Body mass (g) against convex hull volume (mm³). For slope equations see Table 2. Filled circles and solid line, eviscerated carcasses; crosses and dashed line, intact carcasses; dot-dash line, combined sample.

specimens within the sample ($a = -2.31$, $b = 0.90$, $r^2 = 0.97$). In contrast, the relationship between $M_b$ and $CH_{vol}$ in intact specimens illustrates a weak positive correlation characterized by low correlation coefficients and high mean square error ($a = -1.06$, $b = 0.66$, $r^2 = 0.70$). Intact specimens do not plot consistently above the eviscerated pigeon slope (Fig. 2) and are instead characterized by a high degree of scatter. When combining the eviscerated and intact specimens into one dataset, $M_b$ and $CH_{vol}$ remain tightly correlated ($a = -2.08$, $b = 0.85$, $r^2 = 0.92$).

Total $CH_{vol}$ calculated for the mounted dodo skeletons are reported in Table 3 (see Supplemental Information 1 for segment-specific values) and an example of a photogrammetric model is illustrated in Fig. 3. Using the 'eviscerated' predictive model, dressed $M_b$ isestimated as 8.0 kg (95% prediction interval (PI) 4.6–13.9 kg), 8.7 kg (95% PI 5.0–15.0 kg) and 10.8 kg (95% PI 6.1–19.0 kg) respectively for the NHMUK Tring, NHMUK South Kensington and Edinburgh dodos. Applying the 'combined' predictive equation results in wider and therefore more conservative prediction intervals (NHMUK Tring, 6.7 kg 95% PI 3.5–13.1 kg; NHMUK South Kensington, 7.3 kg 95% PI 3.7–14.3 kg; Edinburgh, 9.0 kg 95% PI 4.5–17.9 kg).

The results of the OLS regression of convex hull volume against body mass for a dataset of ground-dwelling ratites and galloanserae derived from *Brassey & Sellers (2014)* are

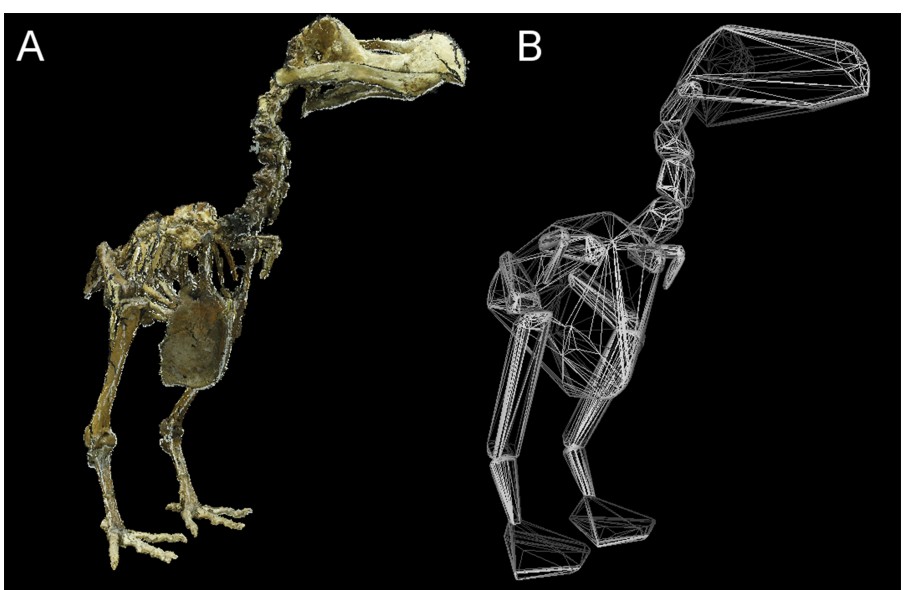

**Figure 3 Convex hull model of Tring dodo.** (A) Photogrammetry model of the Tring dodo skeleton (S/1988.50.1); (B) volumetric convex hulls fitted around the skeleton.

**Table 3 Predicted body mass of the dodo.** $M_b$ estimated using 'eviscerated' equation minus feet (Table 2) and applying correction factor $e$(MSE/2) to account for back-transformation of a log-linear model into a power function, where MSE is the mean square error reported in Table 2. Ground-dwelling refers to the predictive equation based upon ratites and fowl derived from *Brassey & Sellers (2014)*.

| Model | Accession number | $CH_{vol}$ (mm$^3$) | $CH_{vol}$—feet (mm$^3$) | Eviscerated $M_b$ (g) | 95% PI (g) | Ground-dwelling $M_b$ (g)[a] | 95% PI (g) |
|---|---|---|---|---|---|---|---|
| NHMUK Tring dodo | S/1988.50.1 | 8,942,820 | 8,445,134 | 7,980 | 4,653–13,685 | 10,869 | 5,737–20,593 |
| NHMUK Kensington dodo | NHM A.9040 | 9,730,367 | 9,283,795 | 8,687 | 5,027–15,011 | 11,646 | 6141–22,084 |
| Edinburgh dodo | NMS.Z.1993.13 | 12,147,000 | 11,787,000 | 10,760 | 6,106–18,961 | 13,960 | 7,338–26,560 |

**Notes.**
95% PI, 95% prediction intervals..
[a] Calculated on the basis of dodo $CH$ vol including feet, as per the modern ground-dwelling birds

presented in Table 2. This relationship is also characterized by high correlation coefficients ($a = -1.65$, $b = 0.82$, $r^2 = 0.97$), and results in *intact* mass estimates of 10.9 kg (95% PI 5.7–20.6 kg), 11.6 kg (95% PI 6.1–22.1 kg) and 14.0 kg (95% PI 7.3–26.6 kg) respectively for the NHMUK Tring, NHMUK South Kensington and Edinburgh dodos.

Figure 4 illustrates the distribution of segment-specific convex hull volumes as a proportion of total $CH_{vol}$ within the models. In extant pigeons trunk $CH_{vol}$ represents on average 69% of total $CH_{vol}$. The NHMUK Tring dodo skeleton has a percentage trunk volume significantly lower than that of extant pigeons (67%, 1-tailed $t$-test, $t = 3.23$, $p < 0.01$), whilst percentage trunk volume in the NHMUK South Kensington and Edinburgh skeletons is significantly higher than extant pigeons (71% and 80%, $t = -2.23$ and $-13.0$ respectively, $p < 0.05$). With the exception of the tarsometatarsii of the NHMUK South Kensington skeleton, pelvic convex hull segments of the dodos comprise a significantly greater proportion of total $CH_{vol}$ than in extant pigeons ($p < 0.05$). In

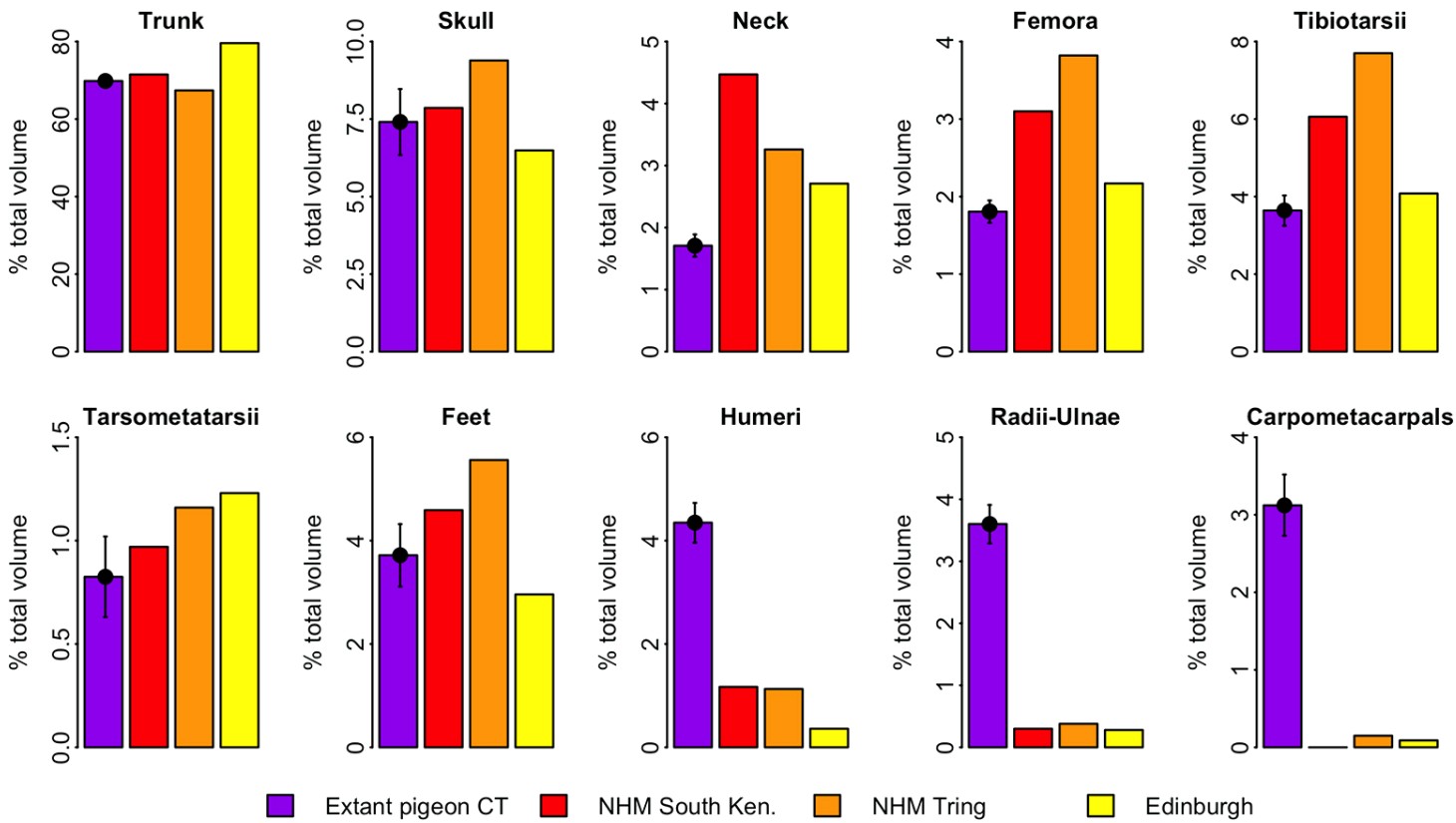

**Figure 4** **Distribution of convex hull volume around the pigeon skeleton.** The distribution of segment $CH_{vol}$ as a proportion of total $CH_{vol}$ within the convex hulled skeletons of extant pigeons and articulated dodo skeletons. Mean values are illustrated for extant pigeons. Error bars represent 95% confidence intervals of the mean. The underdevelopment of the pectoral girdle (humerus, radius and ulna and carpometacarpals) in dodo relative to extant pigeons is particularly striking.

contrast, dodo pectoral convex hull segments contribute proportionally less to total $CH_{vol}$ than in extant pigeons ($p < 0.0001$) (Fig. 4).

## DISCUSSION

### Predictive equation derived from modern CT dataset

To our knowledge the present study represents the first application of a predictive equation derived solely from whole-body CT to the problem of body mass estimation for extinct animals. Previous volumetric mass estimate studies have relied upon articulated museum skeletons of extant species to derive a calibration equation (*Brassey et al., 2013*; *Sellers et al., 2012*). Yet articulated skeletons are often missing crucial specimen information, such as a recorded body mass. By working with frozen carcasses, body mass is directly measurable and uncertainties associated with mounting and posing of the skeletons can be avoided (*Brassey & Sellers, 2014*).

Our dataset consists of both 'intact' and 'eviscerated' pigeons as determined by examination of CT scans. Previous analyses of carcass composition have found eviscerated mass to represent 62–66% of live body mass in rock doves (*Ibrahim & Bashrat, 2009*; *Omojola, Isa & Jibir, 2012*), yet no data exist regarding the possible scaling of internal organ

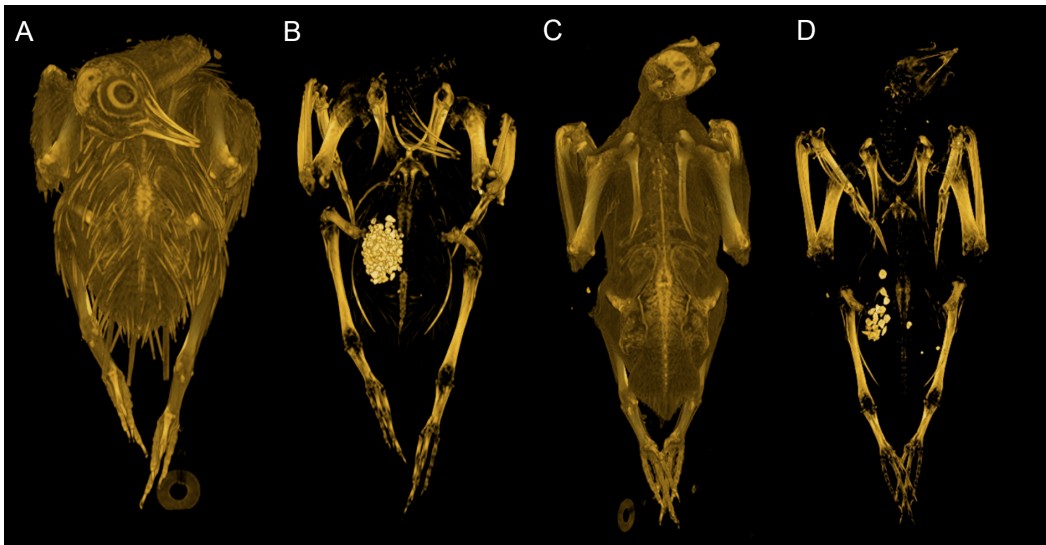

**Figure 5 Volume renderings of modern pigeon CT data.** Volumetric renderings of a rock dove (*Columba livia*, (A–B)) and collared dove (*Streptopelia decaocto*, (C–D)) generated from CT scans. (A) and (C) illustrate the outer soft tissue contours of the carcass, while (B) and (D) illustrate the position of the gizzard and associated gizzard contents. There is considerable variation in the quantity and size of gizzard stones between intact pigeon specimens within the dataset. Renderings were generated in OsiriX (*Rosset, Spadola & Ratib, 2004*).

mass across a range of body sizes in the Columbiformes. As can be seen in Fig. 2, there is no consistent disparity between intact and eviscerated specimens, and the relationship between $M_b$ and $CH_{vol}$ in intact pigeons is relatively weak ($r^2 = 70$, $p = 0.019$). This correlation improves considerably when accounting for phylogeny (Supplemental Information 1), but remains weaker than the relationship between $M_b$ and $CH_{vol}$ characterizing eviscerated specimens. Live body mass has been shown to vary considerably in wild animals due to hydration, nutrition and gut content (*Roth, 1990*) and therefore some degree of scatter is to be expected in intact carcasses. Particularly striking is the variability in gizzard contents between similar-sized specimens visible in CT scans (see Fig. 5).

This suggests intact pigeon $M_b$ cannot be corrected for the presence of internal organs using a single factor representing average percentage eviscerated mass as a function of live mass (i.e., multiplying by values of 0.62 or 0.66 previously found in the literature). Additionally, attempting to correct intact $M_b$ by substituting intact $CH_{vol}$ into the eviscerated regression model would be highly circular and result in artificially inflated correlation coefficients, if the equation were used in a predictive capacity. Therefore, we apply both the uncorrected OLS eviscerated model and combined (eviscerated and intact) model to bracket the range of likely dodo body masses. Interestingly, the very high correlation coefficient and low mean squared error of the eviscerated equation suggest that once the variability associated with fluid and gut content is removed, the relationship between the mass of the remaining musculoskeletal system and $CH_{vol}$ is more tightly constrained.

## Volumetric body mass estimation applied to the dodo

No reliable records of the body mass of dodo exist prior to its extinction in the 17th Century and subsequent mass estimates have varied considerably. Early accounts of the flightless bird suggested an average mass of 50 lb (22 kg) (*Herbert, 1634*), although such accounts "have a tendency towards exaggeration" (*Hume, 2006*). More recently a 'slim' dodo (mean 10.2 kg) was proposed on the basis of femoral, tibiotarsal and tarsometatarsal length scaling in modern birds (*Angst, Buffetaut & Abourachid, 2011a*). However, hind limb bone length has been shown to correlate poorly with body mass relative to other cross-sectional geometric properties and frequently contains a strong functional signal (*Field et al., 2013*; *Kirkwood et al., 1989*; *Campbell & Marcus, 1992*; *Brassey et al., 2013*). Alternatively, a predictive equation based on femoral and tibiotarsal least circumference in ground-dwelling birds has suggested mass estimates between 9.5–12.3 kg (*Louchart & Mourer-Chauviré, 2011*).

The application of volumetric mass estimation techniques to the dodo has been rare. A sculpted scale model of a 'slim' dodo based upon mean skeletal measures was created to replicate sketches dating contemporaneously to its survival on Mauritius and resulted in mass estimated of 12.5–16.1 kg (*Kitchener, 1993*). In the same study, a 'fat' dodo model based on later 'exaggerated' artworks was predicted to weigh between 21.7–27.8 kg.

Here we estimate mean eviscerated body masses for articulated composite dodo skeletons of between 8.0–10.8 kg. Without further information regarding the effect of within-subject variability in gizzard, crop or gut content or interspecific scaling of viscera mass, any extrapolation to a live mass should be treated with caution. However, with this caveat in mind, a 33% increase in mass to account for missing organs (as quantified in extant *C. livia*) would take our results to 10.6–14.3 kg. This overlaps with the slim sculpted model based on contemporaneous accounts (*Kitchener, 1993*). Including our 95% prediction intervals takes both the NHMUK Tring and South Kensington skeletons to a maximum of 18.2 kg and 19.9 kg whole body masses, still considerably below the 22 kg suggested historically (*Herbert, 1634*). In contrast, the 95% prediction intervals of the Edinburgh dodo include 22 kg once multiplied by 1.33.

Unlike all previous volumetric studies, our convex hulling technique does not require a value for body density to be assigned from the literature. Instead, we directly derive the relationship between $M_b$ and $CH_{vol}$ in order to avoid uncertainty regarding assigning literature values, which have been shown to differ considerably across avian groups and with various methodologies for estimating body density (*Seebacher, 2001*). However, this does implicitly rely upon the predictive equation being applied to a fossil of an extinct species that is closely related to (and can therefore be assumed to share a similar body density to) the modern dataset from which the predictive equation was derived. This would include soft tissue density, integument density and skeleton density. In this case of estimating dodo mass based on extant pigeons, we believe this assumption can be upheld. In a micro-CT study of femoral and tibiotarsal mid-shaft cross-sectional geometry, the dodo and solitaire have been found to possess limb bone pneumasticity within the range of extant ground-dwelling and flighted species (*Brassey et al., 2013*) for example.

Alternatively, $CH_{vol}$ may be multiplied by a given value of carcass density to give a hard lower limit to body mass (as carcass volume cannot be less than convex hull volume). The sole literature value for intact feathered pigeon density is 648 kg/m$^3$ from *Hamershock, Seamans & Bernhard (1993)*, producing hard lower bounds to estimated body mass 5.8 kg, 6.3 kg and 7.9 kg for the Tring, Kensington and Edinburgh composite skeletons respectively.

We consider the convex hulling technique to be superior to other sculpting-based volumetric methods (such as manual sculpting with clay (*Kitchener, 1993*) or digital sculpting with non-uniform rational B-spline (NURBs) curves (*Bates et al., 2009*)) for the purpose of mass estimation as soft tissues and hypothesized respiratory systems need not be reconstructed for fossils of extinct species, and the technique is entirely repeatable. When values for centre of mass (COM) and segment inertial properties are required for further biomechanical analyses, NURBs may be required in order to achieve a representative mass distribution across the skeleton. In such situations, it is essential that soft tissue reconstructions are based on quantitative comparative dissection data from relevant modern species in order to minimize subjectivity in model creation. However, for the sole purpose of mass estimation, convex hulling should be the preferred technique.

Previous authors have cautioned over the extrapolation of regression models beyond the limits of the extant dataset when applied in a predictive capacity (*Henderson, 2006*). To avoid this scenario, here we also apply a convex hull predictive model previously derived from ratites and ground-dwelling galloanserae birds (*Brassey & Sellers, 2014*) to the mounted dodo specimens. This results in mass estimates for the *intact* dodo ranging between 10.8–14.0 kg, remarkably similar to those values tentatively reconstructed by correcting the eviscerated pigeon model for missing viscera content. This further strengthens the argument for the reconstruction of a relatively slim dodo, and suggests extrapolation of the predictive equation beyond the range of modern pigeons does not, *in this instance*, result in implausible mass estimates.

Yet a predictive equation based upon cursorial ground-dwelling birds might also be considered inappropriate in light of the commonly-held perception of the dodo as being poor at locomotion, i.e., non-cursorial. The issue faced when assembling a modern calibration dataset on the basis of functional/behavioral similarities (as opposed to phylogenetic relatedness) is the requirement to assume a particular function/behaviour in a fossil species. In the case of the dodo, several 'first-hand' descriptions attest to the 'tameness' and 'edibility' of the bird (*Parish, 2013*, and references therein), yet very limited (and contradictory) accounts exist regarding its locomotor performance. Whilst some confirm the perception of dodo as fat and waddling:

> "… her body is round and extremely fat, her slow pace begets that corpulence" (*Herbert, 1638*, p 347)

Others suggest the dodo was capable of fast and 'jaunty' locomotion:

> "they showed themselves to us with an abrupt stern face and wide open mouth, very jaunty and audacious of gait" (*Servaas van Rooijen, 1887*, p 6)

"'[they] could not fly, (because they [had] in place of the wings only small *Flittige*) however [they] run fast" (*Olearius, 1696*, p 152)

In light of this confusion, a more appropriate modern calibration dataset might therefore be selected on the basis of perceived evolutionary pressures (or lack thereof) to which the dodo was subjected, rather than assumed locomotor ability. Yet this also proves problematic, as the fates of many other recent flightless bird species that have evolved in the absence of native terrestrial predators have followed that of the dodo i.e., recent extinction. Possible extant candidates are limited to the kakapo (*Strigops habroptilus*), Galapagos cormorant (*Phalacrocorax harrisi*), Auckland Islands and Campbell Islands teal *(Anas aucklandica* and *Anas nesiotis*) and over a dozen species of Gruiformes. Given that the majority of the above species are categorized as threatened or extinct in the wild (*IUCN, 2013*), obtaining specimens and associated mass data is extremely challenging. Therefore whilst the dodo may have differed from our modern calibration datasets in being both flightless *and* non-cursorial, it must be recognized that a panacea for dodo mass estimation is unlikely to exist, and perhaps the most appropriate recent analogues are already extinct or nearing extinction.

## Composite and articulated skeletons

The dodo specimens included in this study are composite skeletons, comprising skeletal material from more than one individual and including sculpted or cast elements. Therefore, our study is limited to estimating the body mass of the hypothetical animal represented by each articulated skeleton, rather than a known individual. Currently there exists only one near-complete dodo skeleton comprising a single individual (the Thirioux dodo), upon which research is currently still continuing (*Randall et al., 2014*).

Whilst the use of composite skeletons should clearly be treated with caution when used in biomechanical analyses, their composite nature does not entirely rule out their use, particularly in the case of mass estimation. A recent large-scale macroevolutionary (*Benson et al., 2014*) study of body size in a fossil lineage relied upon mass data derived solely from humeral and femoral circumferences of one individual specimen per species. While this approach is often the only one feasible given the highly fragmentary nature of the fossil record, taking one individual as being representative of an entire species leaves us vulnerable to the possibility of high levels of intraspecific variation.

In contrast, a volumetric reconstruction based on a composite skeleton may be more likely to reflect a species average by virtue of being a combination of several individuals and could be less skewed by isolated robust or gracile elements. If subsequent biomechanical analyses are to be carried out (such as finite element analysis on a particular musculoskeletal unit), then it is important that the body mass entered into the analyses is representative of that specific individual. However, for the case of volumetric body mass estimation alone, it ought to be possible to derive a representative species mean from a composite skeleton.

Of more concern is the frequency of missing, deformed or reconstructed material within a fossil mount. Known issues with the dodo mounts included in this study include missing ribs (Edinburgh skeleton), missing carpals (NHMUK South Kensington skeleton),

deformation of the fragile pubis (NHMUK South Kensington skeleton) or the loss of the most of the ischium, pubis and caudal vertebrae (NHMUK Tring skeleton). For a given object, the extent of the convex hull fitted to that object is dictated solely by its geometric extremes. In many ways this is advantageous for volumetric fossil reconstructions as damage occurring within the bounds of the convex hull does not affect our volume estimate. However, when extremities are missing (such as the caudal tip of the pubis), the shape and volume of the convex hull are strongly affected. This is evident in the low percentage trunk volume of the NHMUK Tring skeleton (Fig. 5) compared to those of extant pigeons and other dodos.

Whilst some evidence of underdevelopment of pectoral elements and overdevelopment of pelvic elements in the dodo is discernable relative to extant volant pigeons, Fig. 5 predominantly illustrates the important contribution of trunk volume to total mass estimates. The Edinburgh skeleton has a proportionally more voluminous trunk than that of extant pigeons and other dodo skeletons, and therefore all other skeletal elements contribute proportionally less to total $CH_{vol}$. The more voluminous trunk relative to other specimens may be attributed to the anterior positioning of the sternum due to constraints associated with the armature supporting the mount. The opposite is true of the NHMUK Tring skeleton, in which damage to the extremities of the pelvic girdle result in a reduced trunk volume. This highlights the sensitivity of volumetric reconstructions of fossils of extinct species to trunk morphology, and should be a concern when working with both composite and complete fossil specimens. Whilst the inclusion of cast and/or sculpted material outside of the trunk may also introduce additional uncertainty into the reconstruction, it is unlikely to impact heavily upon mass estimates given their relatively minor contribution to overall volume (Fig. 4).

### Summary

Here we present the first volumetric reconstruction of fossil body mass based entirely on modern whole-animal CT data. The eviscerated body mass of three articulated composite dodo skeletons is estimated to fall between 8.0–10.8 kg. When accounting for missing organ mass, our mean values still fall towards the lower range of previously published mass estimates. As the availability and cost of CT improves, we believe this non-subjective convex hull approach will become increasingly commonplace. Mass estimation of extinct species from fossils relies upon two key components; a reliable calibration equation derived from extant species, and an accurate reconstruction of the extinct individual from its fossil. We discuss the issues surrounding the use of articulated composite skeletons, and highlight the particular importance of trunk morphology to volume reconstructions. We suggest future efforts should focus on quantifying ribcage and sternal geometry in extant groups in order to bracket the possible trunk shape in fossils of extinct species.

## ACKNOWLEDGEMENTS

We thank Zoller+Fröhlich (Z+F) Limited for their continued support, Jennifer Anné and Dr Victoria Egerton (University of Manchester), Martin Baker (University of Liverpool), Malgosia Nowak-Kemp (Oxford Museum of Natural History), Judith White and Sandra

Chapman (Natural History Museum, London), and the reviews of Heinrich Mallison, Julian Hulme and two anonymous referees.

### Funding

The authors received no funding for this work.

### Competing Interests

The authors declare there are no competing interests.

### Author Contributions

- Charlotte A. Brassey conceived and designed the experiments, performed the experiments, analyzed the data, wrote the paper, prepared figures and/or tables, reviewed drafts of the paper.
- Thomas G. O'Mahoney performed the experiments, reviewed drafts of the paper.
- Andrew C. Kitchener contributed reagents/materials/analysis tools, reviewed drafts of the paper.
- Phillip L. Manning performed the experiments, contributed reagents/materials/analysis tools, reviewed drafts of the paper.
- William I. Sellers conceived and designed the experiments, performed the experiments, reviewed drafts of the paper.

### Data Availability

Modern pigeon convex hull data is available via the website animalsimulation.org. Photogrammetry models of dodo are available by request from Dr Andrew Kitchener (National Museum of Scotland; a.kitchener@nms.ac.uk), Sandra Chapman (Natural History Museum, South Kensington; s.chapman@nhm.ac.uk) and Judith White (Natural History Museum, Tring; judith.white@nhm.ac.uk).

### Supplemental Information

Supplemental information for this article can be found online at http://dx.doi.org/10.7717/peerj.1432#supplemental-information.

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
