# Peer review of "Convex-hull mass estimates of the dodo (Raphus cucullatus): application of a CT-based mass estimation technique"

_PeerJ, doi:10.7717/peerj.1432_

## Round 0.1 · original submission · Major Revisions

Dear authors,

The two reviewers have made very different recommendations. As such it has been difficult to reach a decision in this case. However, I have decided to give 'major revisions'. Your paper will need to go through another round of peer review. If at that point rejection is again recommended, I may accept that decision (depending on what is raised). Please pay close attention to the recommendations of reviewer two. Although he recommended rejection, he was very complimentary, and has provided a plan of action for you. Reviewer one has also given some good advice which I would recommend heeding.

I have some minor additional comments that the authors should address prior to resubmission (in addition to those made by the reviewers):

1. Authority and date should be provided for each species-level taxon at first mention. Please ensure that the nominal authority is also included in the reference list.
2. The full institutional abbreviation for the Natural History Museum London is: NHMUK.
3. Introduce abbreviations, such as finite element analysis for 'FEA'.

Reviewer 1 ·

Basic reporting

The present contribution is well written, and well justified in a historical context. The article's structure is appropriate, and the figures are well constructed and labeled. I feel that the present article conforms to the basic reporting standards of PeerJ.

Experimental design

The present submission describes original primary research, and addresses a well-defined question. The methods that have been applied represent a new approach to understanding the live body mass of the dodo, and the authors do a good job of justifying the application of this technique to this historically controversial question. However, I feel that the varying nature of the extant specimens used to develop a convex hull-body mass regression has lead to some technical problems with the interpretation of the authors' results. These will be discussed in more detail below. The research conforms with ethical standards.

Validity of the findings

The authors are primarily interested in estimating the live body mass of the dodo. In order to do that, they regress the body mass of live pigeons against convex hull volumes in order to come up with a predictive equation for estimating mass in the dodo on the basis of its convex hull volume. The authors do a good job of establishing that this approach, in theory, should lead to robust, repeatable estimates of body mass in fossil organisms, provided that closely related extant specimens can be studied for the generation of predictive equations.

However, the present contribution is problematic since the majority of the extant specimens in the analysis have been eviscerated, presumably in the process of a necropsy. Therefore, the majority of the extant specimens employed in the generation of body mass-convex hull volume regressions _do not_ actually reflect the live body masses of those pigeons. Since there is no reason why the authors would be interested in estimating the eviscerated body mass of the dodo, the approach employed here is puzzling. The authors attempt to justify their focus on eviscerated specimens by stating that 'the relationship between [body mass] and [convex hull volume] in intact pigeons is relatively weak" (lines 216-216). However, this weak statistical relationship is almost certainly due to the vanishingly small sample size (seven intact specimens) included in the intact body mass-convex hull regression.

Since the question at hand is 'how much did the dodo weigh', not 'how much did the dodo weigh, minus its gut', the authors must come up with a way to effectively regress convex hull volume against the actual body mass of the pigeons in their analysis, not just their eviscerated mass. I think the way forward would be to use literature values of pigeon body masses (perhaps referring to Dunning's 2007 compendium of avian body masses). Although these values will represent species averages, and in themselves will introduce a degree of statistical uncertainty into the analysis, the values for the pigeons in this dataset will undoubtedly be heavier, and therefore more accurate, than the eviscerated masses currently used by the authors in the present contribution. This approach should allow the authors to obtain live body masses for close to all twenty species in the present dataset, which will improve the statistical performance of their live body mass estimates for the dodo. Performing this additional analysis will be trivially simple, since the convex hull volumes of the extant pigeons have already been estimated, and in my opinion will greatly improve the quality of this paper.

Additional comments

My primary recommendations for improvement to this manuscript are presented in the 'Validity of the Findings' section. Some minor comments follow:

Line 262: "Confidence intervals" were referred to earlier as 95% prediction intervals. I believe the authors indeed mean to say prediction intervals here.

Lines 312-318: Although I have no issue with the authors relying, by necessity, on composite skeletal material for the dodo in this study, I feel that they are taking it a bit far in their argument that a composite skeleton is entirely unproblematic in this context. I think the language should be toned down somewhat.

Lines 333-334: I think the statements about heterochrony are poorly developed at present and, in their current form, are irrelevant to the present study.

Lines 241-245: The authors' argument that "hind limb bone length has been shown to correlate poorly with body mass relative to other cross-sectional geometric properies and frequently contains a strong functional signal" is attributed solely to Brassey et al. 2013. In truth several influential studies have explicitly made this point, and I think they should additionally be referenced here. These include These include Kirkwood et al. 1989, Campbell and Marcus 1992, and Field et al. 2013.

Finally, the authors of the present submission have done well to investigate the influence of phylogenetic autocorrelation in their data by performing PGLS regressions. However, I think some additional justification of their approach is needed. The authors went with a composite pigeon phylogeny and assigned uniform branch lengths across this phylogeny. Given that a comprehensive estimate of avian phylogeny exists that incorporates branch lengths estimated from data (Jetz et al. 2012), I am interested in why the authors elected not to perform their PGLS on this tree.

Citations:

Field DJ, Lynner C, Brown C, Darroch SAF (2013) Skeletal Correlates for Body Mass Estimation in Modern and Fossil Flying Birds. PLoS ONE 8(11): e82000. doi:10.1371/journal.pone.0082000

Jetz, W., Thomas, G. H., Joy, J. B., Hartmann, K., & Mooers, A. O. (2012). The global diversity of birds in space and time. Nature, 491(7424), 444-448.

Kirkwood JK, Duignan PJ, Kember NF, Bennett PM, Price DJ (1989) The growth rate of the tarsometatarsus bone in birds. Journal of Zoology 217: 403-416. doi:10.1111/j.1469-7998.1989.tb02498.x.

Longrich NR, Tokaryk T, Field DJ (2011) Mass extinction of birds at the Cretaceous-Paleogene (K-Pg) boundary. Proceedings of the National Academy of Sciences of the USA 108: 15253-15257. doi:10.1073/pnas. 1110395108. PubMed: 21914849.

·

Basic reporting

all fine

Experimental design

all fine

Validity of the findings

At first glance this paper looks excellent. However, there is a huge elephant of a problem in the room.

My problem with the presented work is that the source data for the regression is unsuitable for the use the authors make of it. “It is bad practice to extend a regression beyond the range of the data it was derived from” (Henderson 2006). Here, the authors fail to heed this important warning, and use source data from birds weighing 71 g to 1951 g, with an average of ~343 g. If we omit the two extremes from the calculation, the average is ~269 g. The median of the entire set is even lower at 232 g. The authors use a regression derived from this data set to estimate the weight of a bird that the pervious literature suggests to have weighed somewhere between 35 to 100 times as much. That’s like using data from human growth studies on elephants – doable, but involving a high risk. Error bars of regressions get outrageously big once one leaves the original range, and get worse and worse the further one moves away.

Sometimes, we can obviously not avoid extending a regression quite far. That is the case if we have no suitable organisms to measure, i.e. either none at all, or none that are sufficiently similar to the one we wish to investigate. The latter is the case when we try to estimate sauropod masses. Today, only whales reach similar body sizes, and comparing aquatic mammals to terrestrial archosaurs is likely to give nonsensical results. Therefore, Sellers et al. quite rightly used the largest land animals they could get data for when they first applied their convex hull method to sauropods. The same applies to Henderson’s 2006 paper addressing sauropod masses, and others (Anderson's femur circumference method comes to mind).

If one does not have suitable taxa available, one must turn to others that are as similar as possible. The key here is the term “suitable” – sister taxa are nice to use, but close taxonomic relations alone are not a good criterion for selection. Usually, closely related taxa are also anatomically and ecologically highly similar, the oddball taxa occur in the best of families. If the taxon to be estimated is morphologically and ecologically dissimilar to its sister taxa, it may be more appropriate to use other, less closely related taxa for comparison, as long as they have evolved under similar pressures and are thus better analogues. Basically, weight is a biomechanical factor, and in biomechanics analogies often are better models than homologies.
In the case presented here, the authors restrict their extant data to animals that are very closely related to the dodo, i.e. to Columbidae. This decision may seem reasonable, but extant pigeons are, as noted above, much smaller than the dodo. Additionally, all taxa used to calculate the regression are volant and live in contact with terrestrial predators from which they typically escape by flight. Not only can they all fly, but most of them fly very well, can do so for extended periods of time, and can even start vertically from the ground. The latter activity is a very strenuous one, and places extreme limits on the thrust-to-weight-ratio of the birds. In sum, the sampled taxa are insufficiently large to produce a regression that can be used to estimate dodo weight reasonably, and are also subjected to quite dissimilar evolutionary pressures. Dodos evolved, for an extended period of time, in a practically predator-free island environment, in which the ability to store enough fat to survive dry seasons surely was a stronger selection pressure than the ability to fly – which explains why dodos were flightless! If Raphus cucullatus had no need for flight and stored extreme amounts of fat against dire times, then we would predict it had extremely stout legs and reduced wings. Both predictions are borne out by the anatomy of the animal, but are completely ignored by the authors. The anatomical details suggest, however, that the regression for mass estimation should ideally be calculated from a large sample of flightless birds with little need for escape from predators, as such birds would have evolved under highly similar conditions and thus would be expected to have a more similar bauplan to the dodo than extant pigeons. The extreme weight difference between the largest flying columbid at ~2 kg and the dodo at something between 8 kg and 25 kg alone should have been a warning sign!

Deplorably, there are preciously few such birds around. However, there are some, and there are quite a number of extant non-volant birds, although many of them can move rapidly to escape predators. And some are indeed not just heavier than pigeons and doves, but even heavier than the heaviest published mass estimate for the dodo!

In my opinion, the authors must create a second regression from non-columbid birds, including flightless large birds like emus, ostriches, rheas (ideally, some juveniles of these taxa, too, in the 2 to 30 kg range), some (near) flightless birds including kagus and kiwis and kakapos and various members of Otididae, and – why not? – penguins! The latter typically do not run away from predators, like dodos, and are quite fatty, like dodos.

Such data would not only allow judging the usefulness of the pigeon-based regression and narrowing down dodo mass, but would also be a great help for other researchers, as there are preciously little reliable mass data published.
If the authors are willing to add such a regression and use it to calculate dodo mass, I can heartily recommend publication after another round of reviews. As it currently stands, however, the MS should be rejected, because it currently makes an unsuitable use of the convex hull method, and because its results are mostly meaningless as a consequence.

Let me emphasize that I recommend rejection NOT because there is any flaw in the method per se, or because the data the authors present on the extant birds are flawed. Both are just fine! All it takes is different extant taxa for a new regression. Therefore, I could also call this a "major revision", but I believe the revisions will take too much time. A new submission, which can then take into account the upcoming Dutch monograph, seems more suitable.

Additional comments

This paper was very tough to review, because I though "wrong regression" already when I read the abstract. I've agonized over the decision "revisions" verses "reject" quite a bit. Now the hot potato is with the editor......
You did excellent work overall, but your approach was "a bridge too near", as you failed to go the full way of getting a suitable regression, and "a bridge too far" as you worry a lot about the errors in the pigeon masses, but in proportion to the dodo mass they are (almost) all tiny.

Please do go to the bother of getting more data points for flightless birds! This will become a landmark paper!

---

## Round 0.2 · Minor Revisions

Dear authors,

The two reviewers have again made very differing recommendations, however I have decided to give 'minor revisions'. Please pay close attention to the recommendations of reviewer two. Although he recommended major revision, he was complimentary on the improvements from the initial submission, and the recommendations suggested would benefit your manuscript.

Reviewer 1 ·

Basic reporting

The paper satisfies these requirements.

Experimental design

The paper satisfies these requirements.

Validity of the findings

The authors have improved the manuscript substantially in response to the comments of two reviewers, and I recommend it for publication.

·

Basic reporting

OK

Experimental design

OK - but still no birds that do not need to run or fly away

Validity of the findings

This MS has improved a lot! It is now in the state I would have expected at first submission, with the author having added a calculation from a regression that comprises "ratites and galloanserae birds" (line 176), i.e. a regression from animals much more similar to dodos than the pigeons and doves that form the basis of the other regressions.
Better - but checking out source [23] shows that the regression is based on ratites and one goose and one guineafowl, both of them much smaller than the suggested dodo mass. And in fact, [23] notes that "the smaller ratites (Apteryx australis, Apteryx australis lawryi) and neognaths possess apparent densities greatly in excess of those predicted for plucked carcasses. This suggests that the volume of ‘missing’ soft tissue located outside the convex hull is greater in smaller birds."
So, four data points are taken in source [23] as a separate group and found to differ from the larger ratites. Now all that data is pooled and supposed to deliver a meaningful result? Maybe.... maybe not.

Also, the authors have not added data from the birds I suggested (and note that I did NOT only suggest penguins) that are not adapted for either flight or speedy running. Geese are excellent fliers if they want to, and source [23] lists only Branta leucopsis. That beast migrates.
The other neognathe in source [23] is Numdia meleagris, which, according to wikipedia, uses "a short-lived explosive flight" like other fowl. Not really suitable tests, just like the fast-running big ratites, for a potentially fat, waddling bird. Potentially - but isn't exactly that the question here? Do the authors want to show us what a "regular bird dodo" would have weighed, or do they want to find out what the actual dodo may have weighed? If the latter is the case, then the data used is insufficient.

Thus, my suggestion of penguins and kakapos etc. wasn't intended to just create work for the authors. Rather, these birds are the closest (I know of) to the potential and to-be-tested fat dodo from the literature. In their reply the authors dismiss this, because they "intuitively" understand that penguin density is different - but hey, is kakapo density different? And volume and density are separate paramenters (this distinction being ignored was already a problem in the original paper introducing the convex hull method). If one wants to, one can measure penguin volume and apply an average non-penguin density to derive a hypothetical terrestrial penguin.

By the way, I accept that kakapo and kagu data may be hard to get (have the authors tried?). Penguins are in freezers all over the zoos of the world - and at RVC (JR Hutchinson). That wouldn't have been too hard to get.

So what to do now? The authors made progress, and the MS is now in a state where I feel the data and conclusions can be published. And I do NOT want to demand data that would really be a huge bother to get.

Thus, I suggest the paper gets provisionally accepted. Provided, that is, that the authors DISCUSS the problems that remain, detailing how possibly(!) dodos were unusual due to the lack of evolutionary pressure for rapid escape (flight or running). If they do not do these minor in work effort but major in character revisions properly, sourcing the literature that suggested exactly this scenario, the paper should be rejected.

---

## Round 0.3 · Minor Revisions

Dear authors,

The reviewer who recommended 'major revisions' in the previous rounds was unavailable for this round of review. As such, I have sent it out to two more reviewers. I have accepted the decision of 'minor review'. However, as the comments and suggestions of both reviewers are fairly minor in nature, your manuscript will not require to undergo a further round of review.

Once again, thank you for submitting your manuscript to PeerJ and I look forward to receiving your revision.

·

Basic reporting

No Comments

Experimental design

No Comments

Validity of the findings

No Comments

Additional comments

Dear Authors,
This is a well written paper and I have little to add to it. I have however made some very minor edits and suggestions; otherwise the paper is ready for publication. My edits and suggestions are in track changes on the word document.

Reviewer 4 ·

Basic reporting

No comments

Experimental design

I think that the data design is fine here. My only question is, if the authors have checked that the amount of postcranial intraosseous pneumaticity is comparable with that of the doves they used - if not, and the Raphus bones are less pneumatic, i.e., more massive, it might change the calculation of the body mass. I feel that it might be useful to clarify this, but only with a short sentence in the Materials and Methods section, so it would neither be necessary to re-review the paper, nor to expand the provided data or images in the paper.

Validity of the findings

No comments

Additional comments

I found the paper a comprehensive study and a very sophisticated approach to approach the body mass of the Dodo. All possible arguments are discussed and the calculations are convincing to me. I have therefore no comments for changes, except the one outlined above, but I think that this does not hinder acceptance of the paper.

---

## Round 0.4 · accepted · Accept

Dear authors,

Thank you for swiftly responing to the reviewers comments. I am delighted to inform you that your manuscript has been accepted for publication.